# Positive Association between Peri-Surgical Opioid Exposure and Post-Discharge Opioid-Related Outcomes

**DOI:** 10.3390/healthcare11010115

**Published:** 2022-12-30

**Authors:** Kibum Kim, Joseph E. Biskupiak, Jennifer L. Babin, Sabrina Ilham

**Affiliations:** 1Department of Pharmacy Systems, Outcomes and Policy, University of Illinois at Chicago, Chicago, IL 60564, USA; 2Department of Pharmacotherapy, University of Utah Health, Salt Lake City, UT 84112, USA; 3Pharmacotherapy Outcomes Research Center, University of Utah Health, Salt Lake City, UT 84112, USA

**Keywords:** persistent opioid use, opioid related adverse event, healthcare resource utilization, surgery

## Abstract

Background: Multiple studies have investigated the epidemic of persistent opioid use as a common postsurgical complication. However, there exists a knowledge gap in the association between the level of opioid exposure in the peri-surgical setting and post-discharge adverse outcomes to patients and healthcare settings. We analyzed the association between peri-surgical opioid exposure use and post-discharge outcomes, including persistent postsurgical opioid prescription, opioid-related symptoms (ORS), and healthcare resource utilization (HCRU). Methods: A retrospective cohort study included patients undergoing cesarean delivery, hysterectomy, spine surgery, total hip arthroplasty, or total knee arthroplasty in an academic healthcare system between January 2015 and June 2018. Peri-surgical opioid exposure was converted into morphine milligram equivalents (MME), then grouped into two categories: high (>median MME of each surgery cohort) or low (≤median MME of each surgery cohort) MME groups. The rates of persistent opioid use 30 and 90 days after discharge were compared using logistic regression. Secondary outcomes, including ORS and HCRU during the 180-day follow-up, were descriptively compared between the high and low MME groups. Results: The odds ratios (95% CI) of high vs. low MME for persistent opioid use after 30 and 90 days of discharge were 1.38 (1.24–1.54) and 1.41 (1.24–1.61), respectively. The proportion of patients with one or more ORS diagnoses was greater among the high-MME group than the low-MME group (27.2% vs. 21.2%, *p* < 0.01). High vs. low MME was positively associated with the rate of inpatient admission, emergency department admissions, and outpatient visits. Conclusions: Greater peri-surgical opioid exposure correlates with a statistically and clinically significant increase in post-discharge adverse opioid-related outcomes. The study findings warrant intensive monitoring for patients receiving greater peri-surgical opioid exposure.

## 1. Introduction

Pain management is an integral part of patient care after surgery. Certain major surgeries, including spine surgery (SS) and cesarean delivery (CD), are rated among the most painful procedures [1,2], and patients often experience moderate-to-severe pain after frequently performed orthopedic surgeries and hysterectomy (uterus removal, UR) [3,4,5,6]. In a large retrospective analysis of patients who underwent total hip arthroplasty (THA), 8% and 11% of patients experienced moderate-to-severe pain even after they passed multiple years, respectively [7]. Similarly, ~78% of women who undergo CD experience moderate-to-severe pain after surgery [2].

Pain control after surgery often involves opioid regimens that are intended primarily as short-term management of “breakthrough pain” [3,8,9]. Although opioids are effective analgesics, they are associated with the occurrence of opioid-related symptoms (ORS), the development of opioid use disorder, and overdose [10,11,12,13]. Exposure to opioids can lead to an increase in healthcare resource utilization (HCRU), such as prolonged hospital length of stay (LOS), elevated readmission rates, and increased overall healthcare costs [12]. Given these concerns, the American Pain Society, the American Society of Regional Anesthesia and Pain Medicine, and the American Society of Anesthesiologists jointly developed a set of guidelines in which a multimodal pain management is recommended [14]. The guidelines emphasized that systemic opioid use may not be required in all patients [14].

Despite guidance recommending the carefully planned prescription of opioids, patients who receive surgical intervention often use opioids for long-term pain management and develop opioid use disorder or ORS [12,15,16,17,18,19,20]. A cross-sectional study demonstrated that the likelihood of opioid use disorder in patients hospitalized for spinal conditions increased by 5.2% annually from 2005 to 2014 [21], and more than half of SS patients remained chronic users through 12 months [16]. Regarding the studies in orthopedic surgery, about 14% of Australian patients used opioids for longer than 90 days after surgery [20]. Approximately 10% of the orthopedic operations were followed by post-discharge opioid use for longer than a year [22]. When the analysis was limited to obstetrics and gynecology (OBGY), approximately 85% of patients filled an opioid prescription after CD, and 67% filled one after minimally invasive UR [19,23,24]. Considering a large number of opioids are left unused, overprescribing opioids may also lead to increased opioid consumption without improving patient satisfaction or proper pain management [19,25,26].

Multiple studies have shown that patients who use opioids in the pre- and peri-operative setting are more likely to use opioids after discharge, with a resulting significantly higher rate of ORS than opioid-naive patients [26,27,28]. Nevertheless, the adverse influence of peri-surgical opioid use on post-discharge opioid-related outcomes such as persistent opioid prescription, ORS, or increase in HCRU remains unclear.

The objective of this retrospective analysis was to narrow the knowledge gap by (1) describing the prescription of opioid medications for post-operational pain management and (2) testing the association between the level of opioid exposure during inpatient admission for frequently performed surgical interventions, including SS, THA, total knee arthroplasty (TKA), CD, and UR and persistent post-discharge opioid prescription. This research also aims to provide insight into the safety and economic concerns associated with peri-surgical opioid use by evaluating the diagnosis of ORS and comparing post-discharge HCRU across the varying levels of perioperative opioid exposure.

## 2. Methods

### 2.1. Study Design and Patient Population

This was a retrospective cohort analysis of administrative data and electronic health records from the Enterprise Data Warehouse of the University of Utah Healthcare System. The database contains patient-level diagnoses, procedures, ordered and/or prescribed medications, administrative claims, and detailed encounters. This study was deemed exempt from institutional review board review on 27 January 2017.

The analytic cohort included male and female patients who were aged >18 years when they received one of the index procedures including SS, THA, TKA, CD, and UR between January 2015 and June 2018. Surgeries were defined by the *International Classification of Diseases, Ninth and Tenth Editions*, and Current Procedural Terminology procedure codes (Appendix A). Patients were excluded if they were aged <18 years, were pregnant at the time of surgery other than CD, had a history of oncology care, or underwent any procedure that required general anesthesia within the 180-day period before the admission for the index surgery. Eligible subjects had to have at least one follow-up record on or after 180 days after the index discharge to meet the 180-day post-discharge outcome assessment criteria.

Baseline demographics and clinical characteristics during the 180 days prior to and on the index admission were extracted from the data using administrative and treatment records. Patient demographics included age, gender, health plan, race, and ethnicity. Clinical characteristics included weight, body mass index (BMI), baseline opioid use, comorbid conditions defined by the Charlson Comorbidity Index (CCI) [29], history of alcohol or drug use, diagnosis of opioid-related adverse events (ORAEs) from administrative coding, and use of nonopioid prescription pain medications.

Peri-surgical opioid exposure was defined during the inpatient stay for the index surgery. The opioid dose was determined based on the medication administration data including dose per unit (ml/pill/capsule), rate, duration, total volume, and the number of pills or capsules per administration, and it was converted to the morphine milligram equivalent unit (MME, mg) using MME conversion factors (Appendix A). Patients were divided into two groups determined by the surgery group-specific median MME: greater than the median MME vs. at or lower than the median MME.

### 2.2. Outcomes

The primary outcome of this study was persistent opioid use, defined by two criteria during the 180-day post-discharge period. The ‘30-day criterion’ was ≥1 opioid prescription within 30 days after discharge and another opioid prescription(s) between 31 and 180 days after discharge. The ‘90-day criterion’ was ≥1 opioid prescription within 90 days after discharge and another opioid prescription(s) between 91 and 180 days after discharge. Describing the pattern of opioid prescription by peri-surgical opioid exposure, we summarized monthly prescription of opioid medication over the 180-day follow-up period, too.

The secondary outcome of this study was the proportion of patients having a diagnosis of ORAEs during the 180-day post-discharge period. ORAEs were defined using the *International Classification of Diseases* algorithm, as previously described [11]. Rates of ORAEs were measured overall and stratified by type (cardiovascular, central nervous system, gastrointestinal, genitourinary, respiratory, dermatologic, and unspecified). The purpose of this study was to determine any signal in increasing ORAEs associated with index MME, and we did not specifically determine whether the diagnosis of ORAEs was caused by opioid use. HCRU as a secondary outcome was assessed through 30, 60, and 180 days post-discharge, including the number of hospital readmissions, LOS, the number of office visits, and the number of emergency department (ED) visits for any reason.

### 2.3. Statistical Analysis

Both patient characteristics and outcomes were summarized using descriptive statistics, including mean ± standard deviation (SD), median (interquartile range (IQR)), frequency, and percentage. Statistical comparison between the high and low MME groups was performed using student *t* and Wilcoxon rank sum tests for continuous values and a Chi-square test and Fisher’s Exact test for categorical variables. Testing multiple exposure–outcome associations using the same analytic cohort raises concerns regarding type I error. In addition to the unadjusted *p*-values from bivariate analysis, we applied the Holm method (i.e., Bonferroni step-down or Bonferroni–Holm approach) to a set of statistical tests for each outcome group: persistent opioid use (30-day and 90-day criteria), ORAE, and HCRU.

We tested the association between greater MME exposure and the primary outcome using a logistic regression model. The odds ratio of being a persistent opioid user for the high vs. low MME during the index admission was calculated using a multivariable logistic regression model. Using a step-wise model selection process with the *p*-value for the model entry and stay of 0.1, we tested all the variables in the Table 1 and selected the most influential factors as potential confounders. In the final model, the association between the greater exposure and outcome was adjusted for the dissimilar patient baseline characteristics, including race/ ethnicity, baseline opioid use, use of nonopioid pain medication, ORAE diagnosis at baseline, type of health plan, grouped age (>55 years for SS, >35 years for CD, >45 years for UR), body mass index (>30 kg/m^2^), grouped CCI scores (0, 1, 2 or ≥3), and length of index admission > 5 days. All statistical analyses and tests were performed using SAS, version 9.4 (SAS Institute, Cary, NC, USA).

## 3. Results

### 3.1. Baseline Patient Demographics and Clinical Characteristics

A total of 3939 and 3946 patients were respectively classified as high MME and low MME patients (Table 1). Among the eligible patients, 2451 underwent SS (high MME group, *n* = 1225; low MME group, *n* = 1226), 2309 women underwent CD (high MME group, *n* = 1154; low MME group, *n* = 1155), 310 women underwent UR (high MME group, *n* = 155; low MME group, *n* = 155), 1233 patients had THA (high MME group, *n* = 616; low MME group, *n* = 617), and 1582 patients had TKA (high MME group, *n* = 789; low MME group, *n* = 793). The overall median MME at the peri-surgical setting across the five surgery groups was 309 mg and 67 mg for the high and low MME groups, respectively (Table 1). An analysis stratified by surgery group is presented in the supplemental data tables (Appendix A).

In general, the high MME group was younger (50.2 ± 17.1 years vs. 54.1 ± 18.6 years) and stayed longer for the index admission (5.47 ± 6.71 days vs. 3.96± 2.96 days) than those in the low MME group (*p* < 0.0001). A history of smoking (31.7% vs. 24.4%, *p* < 0.01), an opioid prescription at baseline (25.7% vs. 19.5%, *p* < 0.01), and a diagnosis of ORAE (24.5% vs. 19.8%, *p* < 0.01) were more prevalent in high MME patients. Comorbidity profiles were similar, as the distribution of CCI scores did not differ between the high MME and low MME groups (*p* = 0.98). The proportion of general anesthesia used (vs. regional or other types of anesthesia) was higher in the high MME group (54.4% vs. 40.1%, *p* < 0.01, Table 1).

Analysis stratified by the surgical specialty resulted in outputs similar to the overall cohort assessment, but the difference in some patient characteristics became insignificant due to the decrease in the number of patients analyzed. A baseline exposure to opioids and a diagnosis of ORAEs during the baseline period were significantly more prevalent among the high MME group in reference to the low MME patients in the THA, TKA, and CD cohort (*p* < 0.01, Appendix A). In the SS cohort, the proportion of patients reporting opioid use during the baseline period was similar (*p* = 0.23) between the high and low MME groups, as were rates of ORAEs with or without prior opioid exposure (*p* = 0.18, Appendix A).

### 3.2. Persistent Opioid Use

In general, 93.8% of the eligible subjects received prescriptions for opioid medication within 30 days after the index discharge. The proportion of patients who received subsequent opioid prescriptions significantly decreased over time (18.3%, 11.9%, 9.5%, 8.0%, and 8.0%, from the second to the sixth month after the index discharge, respectively), but the rate was consistently higher among the high MME than the low MME group (*p* < 0.01, Appendix A). A higher rate of opioid prescriptions in the high MME compared to the low MME group was universal in the CD, SS, THA, and TKA subgroup analyses, although the CD and SS cohorts lose statistical significance at the fifth and sixth month after the index discharge. A similar trend was observed within the UR cohort at month 1 and month 2 after surgery, but not at any subsequent time point (Appendix A).

Overall, 33.5% of the high MME and 24.4% of the low MME patients were defined as a persistent opioid user by the 30-day criterion. Based on the 90-day criterion, the proportion of high MME and low MME patients classified as persistent opioid users were 30.7% and 13.8%, respectively. The respective adjusted odds ratios for the 30-day and 90-day definitions were 1.38 (95% confidence interval (95CI): 1.24–1.54, *p* < 0.01) and 1.41 (95CI: 1.24–1.61, *p* < 0.01) (Table 2, Figure 1 and Figure 2). The proportion of persistent opioid users was consistently higher among the high MME group across the five index surgery strata, although the results were not statistically significant from some subgroup analyses. (Table 2, Figure 1 and Figure 2).

### 3.3. ORAEs and HCRU

The proportion of patients with a diagnosis of ORAE during the 180-day post-discharge period was significantly larger in the high MME group compared to the low MME group in the overall analytic cohort (27.2% vs. 21.2%, *p* < 0.01) and in almost all surgery type cohorts (SS, 32% vs. 27%, *p* < 0.01; CD, 19.1% vs. 11.9%, *p* < 0.01; UR, 38.1% vs. 30.3%, *p* = 0.15; THA, 25.3% vs. 20.7%, *p* = 0.06; and TKA, 31.4% vs. 24.7%, *p* < 0.01; Table 3), although the results were not statistically significant in the UR and THA patients. The significant difference in the rate of having ORAE codes was largely attributable to cardiovascular (8.3% vs. 5.5%, *p* < 0.01) and gastrointestinal problems (11.0% vs. 8.4%, *p* < 0.01) in overall and surgery specific analyses (Table 3).

On average, high MME patients had 0.2 ± 0.51 (mean ± SD) admissions during the 180-day post-discharge period, whereas low MME patients were admitted 0.13 ± 0.44 times (*p* < 0.0001, Table 4). With a larger number of admissions, cumulative LOS over the 180-day period out of all eligible subjects was also greater in the high MME patients than in the low MME group (1.09 ± 4.75 vs. 0.64 ± 3.56, *p* < 0.0001). Limiting the analysis to the patients who had at least one hospital admission (*n* = 561 vs. 339 from high MME vs. low MME), we found the average length of stay was still longer after greater exposure to peri-surgical opioids (7.7 ± 10.4 vs. 6.3 ± 9.5, *p* < 0.01). Although the differences were small, there were also statistically significant differences in the number of ED visits (0.24 ± 0.86 vs. 0.18 ± 0.91, *p* < 0.01) and outpatient office visits (5.4 ± 6.9 vs. 5.0 ± 6.2, *p* = 0.04) over the 180-day follow-up period, which consistently favored the low MME groups. Further, patients classified as high MME had more telephone encounters (1.3 ± 1.6 vs. 1.1 ± 1.4, *p* < 0.01) within 7 days after the discharge. (Table 5). From the stratified analysis, not all surgery types demonstrated the lower rate of HCRU in the low MME group or demonstrated a statistically significant difference. Nevertheless, the overall trends in the positive association between the increase in peri-surgical opioid use and HCRU were obvious across the surgery groups (Table 4 and Table 5, Appendix A).

### 3.4. Multi-Test Effect on p-Values

The Bonferroni–Holm method has a moderate impact on the statistically significant and calculated *p*-values, demonstrating statistical insignificance in some cases, specifically when the analysis was limited to a small subgroup, such as UR or THA. In the overall-group assessment, statistical inference was not influenced after we addressed the increase in the chance of type I error.

## 4. Discussion

Using comprehensive medical and administrative healthcare records, this study assessed post-discharge opioid prescription, persistent opioid use, and the presence of ORAE and HCRU associated with increased peri-surgical opioid exposure. From this retrospective study, opioid exposure during inpatient admission for frequently performed orthopedic surgeries and OBGY operations was associated with an increase in opioid-related adverse outcomes. A clinically and statistically meaningful difference between the high and low peri-surgical MME groups was observed across the subspecialty areas including SS, CD, THA, and TKA. The numerical findings were also observed in the hysterectomy cohort, although fewer association measures were statistically significant. This was mainly because of the relatively smaller sample size of the hysterectomy cohort (*n* = 310) compared to the other surgery cohorts, generating less precise and wider confidence interval estimates for the measure of association in the hysterectomy patients.

This study demonstrated the strong and significant association between peri-surgical opioid utilization and post-operational persistent opioid use, which is consistent with previous assessments on long-term opioid use triggered by opioid exposure immediately after surgery or in the past. Of the patients receiving SS, 41% to 52% of patients who had previously been exposed to opioids reported post-operational prolonged opioid use, whereas the rate of prolonged-opioid use was 28% out of opioid-naïve SS patients [16,30,31]. Similarly, a recent analysis of patients undergoing THA or TKA showed that preoperative opioid dose was significantly predictive of post-operational opioid use at 6 months post arthroplasty, with an odds ratio of 1.07 [28]. The opioid exposure and chronic post-surgical opioid association was still significant after being adjusted for the baseline risk factors of complicated medical and surgical histories [32], which was reproduced in our analysis. Although the current study did not elaborate on the rationale for chronic opioid use after orthopedic surgery, a potential explanation may be that higher doses can provide relief from affective distress, leading to opioid-induced hyperalgesia or dependence [28].

Women are more likely to use prescription opioids compared to men with respect to pain sensitivity and more chronic conditions causing pain [33,34,35]. Describing the peri-surgical and post-discharge opioid use after OBGY operation, our study provides real-world opioid utilization statistics for a large female population, with a prevalence of hysterectomy of 11% between the ages of 40–44 [36] and CD in one out of three pregnancies [37]. The rates of persistent opioid use after the OGBY surgeries were lower than that of the orthopedic surgeries and generally consistent with previous studies. For example, the rates of 90-day opioid persistence after CD (2% of low MME and 5% of high MME groups) were comparable to a previous study reporting ~2% opioid persistence among women who received peripartum opioid prescriptions [18]. Further, our study confirmed the increase in opioid persistence after hysterectomy associated with peri-hysterectomy opioid use in the US healthcare setting that was evident after elective gynecologic surgery among a Canadian population [38].

Similar to our study findings, previous reports demonstrated that preoperative opioid use is a predictor of increased HCRU [39,40]. For instance, a retrospective economic analysis of post-posterior lumbar fusion found that patients who regularly consumed opioids prior to surgery were 15% more likely to be readmitted to the hospital than patients not continuously taking opioids previously [39]. A recent cost burden study showed a positive correlation between post-surgical HCRU and opioids before surgical operations including general, orthopedic, plastic, and obstetric/gynecologic surgeries [41]. The larger proportion of patients having administrative diagnosis codes for ORAEs after high MME, which also corresponds to the previous study results [40], was a strong signal for the increase in HCRU mediated by the onset of ORAEs. The hypothesis warrants further assessment to evaluate the index MME—ORAE—HCRU associations using structural equation modeling or mediator analysis.

Several attempts have been made to provide a specific recommendation or implementation strategy for the use of opioids in patients receiving common surgical procedures [42,43,44]. According to the John’s Hopkins procedure-specific recommendation, oxycodone 5 mg prescriptions ideally should not exceed 20 tablets for opioid-naïve patients on discharge, which is equivalent to 50 mg IV morphine or 150 mg PO morphine. In our analysis, however, only 27% of the opioid-naïve patients included in this analysis did not exceed this MME limit, and 50% exceeded 2.5 times this recommended range within 30 days after the index discharge, which is generally consistent with a previous study after the implementation of opioid guidelines [44]. The overutilization of opioids among the post-operational population and increase in peri-surgical opioid use followed by less favored clinical outcomes warrants future investigation into guided opioid use prescribing [44]. Considering that the pain immediately after surgery is worrisome among the US population but not as much of a concern in other countries [45,46], surgeons in the US healthcare setting will have further opportunity to decrease the overutilization of opioids via implementing a planned opioid administration strategy during surgery and in the peri-surgical inpatient setting.

The interpretation of our data should be considered in light of several limitations. First, both peri-surgical and post-discharge opioid use would be determined by the type of procedure or the severity of the condition(s), which confound the association measures. Similarly, existing conditions and previous exposure to opioids also influence both exposure and outcomes. To address this limitation, the regression model included comorbid conditions, the presence of ORAE diagnosis codes at baseline, and the length of index inpatient stay as a proxy estimate for severity. However, the multivariable approach might not completely rule out the potential source of bias and confounding effects. Another limitation of this study was the gap between opioid prescription and exposure. Because our research was limited to a single-center medical intervention and medication order data not comprehensively linked with dispensing records, there might exist a sizable gap between the number of prescriptions ordered versus filled. Nevertheless, considering that our estimates are comparable to the previous real-world assessments of persistent opioid use, the study findings will be a significant addition to understanding surgery-related opioid use. Lastly, our findings should be considered in the context of the retrospective observational research design, which is subject to misclassification due to coding, incomplete records, unobserved confounders, and limited generalizability.

Besides the several limitations owing to the research design, the study period should also be carefully considered for any interpretation and future studies. Patients who are in desperate need of orthopedic intervention, obstetric surgery, or gynecologic operation would be similar before and after the COVID pandemic, and therefore, the strong positive association between the peri-surgical opioid use and post-discharge outcomes would be consistently observed. Nevertheless, systemic changes in resource allocation due to the pandemic potentially influenced resource utilization and opioid prescription patterns. Such structural changes might have a nominal-to-moderate influence on the magnitude of the association. Performing any future analyses, investigators should consider extending the study period to cover the COVID epidemic and adjusting for pre-/post-pandemic effects.

Despite these limitations, our study improves insight into surgical opioid prescription and has many strengths. First, our extended length of follow-up (6 months from discharge, partitioned by month) for multiple outcomes enabled comparisons from both a quality measure perspective (e.g., 30-day readmission rates) and from a payer perspective (overall HCRU). Additionally, we included two different criteria to define persistent opioid use that resulted in similar association measures across the two definitions, suggesting that the conclusions of this study are reliable regardless of the specific criterion used. Further, a large sample for most surgery types allowed for appropriate statistical analyses and the obtainment of consistent findings across the surgical specialty areas.

The present analysis found that patients undergoing CD, UR, SS, THA, and TKA who received high levels of peri-surgical opioid exposure had a greater risk of postsurgical persistent opioid use, more ORAEs, and increased HCRU than those with low opioid exposure, although further causality assessment is warranted. The findings provide insight into the adoption of optimal multimodal pain management strategies that reduce the use of peri-surgical opioids and may help reduce persistent opioid use and HCRU.

## Figures and Tables

**Figure 1 healthcare-11-00115-f001:**
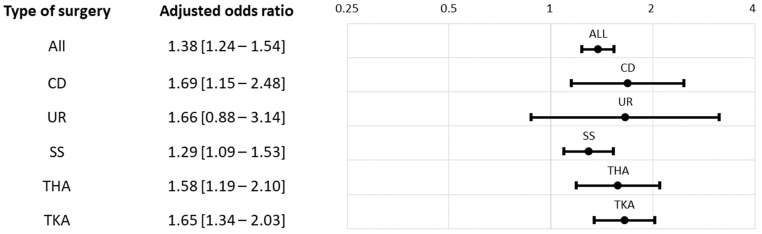
Adjusted odds ratio of post-30-day opioid Rx for index high MME vs. low MME. Abbreviations: CD, cesarean delivery; THA, total hip arthroplasty; TKA, total knee arthroplasty; SS, spine surgery; UR, hysterectomy (uterus removal).

**Figure 2 healthcare-11-00115-f002:**
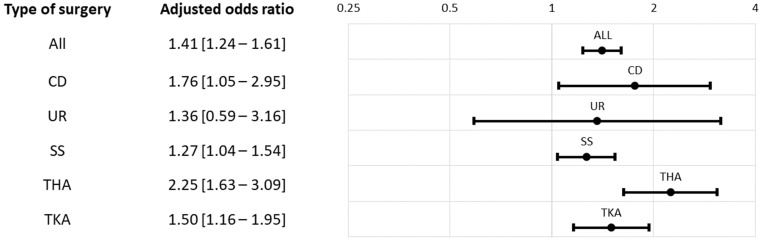
Adjusted odds ratio of post-90-day opioid Rx for index high MME vs. low MME. Abbreviations: CD, cesarean delivery; THA, total hip arthroplasty; TKA, total knee arthroplasty; SS, spine surgery; UR, hysterectomy (uterus removal).

**Table 1 healthcare-11-00115-t001:** Patient characteristics during the baseline (180 days before the index admission if not specified elsewhere) and index admission: high vs. low inpatient opioid exposure.

	High MME	Low MME	*p*-Value
N	3939	3946	
MME (mg), mean (SD)	393.49 (3466.59)	67.38 (70.19)	<0.0001
MME (mg), median	177.02	41.64	<0.0001
MME (mg), IQR	92.70–400.25	23.45–66.67	
Index LOS (day), mean (SD)	5.47 (6.71)	3.96 (2.96)	<0.0001
MME per day, mean (SD)	64.86 (72.05)	20.15 (25.70)	<0.0001
MME per day, median	40.72	12.26	<0.0001
MME per day, IQR	21.47–90.49	7.00–21.59	
Age (year), mean (SD)	50.18 (17.07)	54.09 (18.56)	<0.0001
Height (cm), mean (SD)	168.22 (10.55)	167.74 (10.66)	0.0599
Weight (kg), mean (SD)	89.17 (22.04)	84.69 (20.90)	<0.0001
BMI (kg/m^2^), mean (SD)	31.46 (7.01)	30.07 (7.15)	<0.0001
Year-2015, N (%)	1172 (29.7%)	878 (22.3%)	<0.0001
Year-2016, N (%)	1173 (29.7%)	1033 (26.2%)	
Year-2017, N (%)	1107 (28.1%)	1382 (35.1%)	
Year-2018, N (%)	487 (12.3%)	653 (16.6%)	
Female gender, N (%)	2733 (69.3%)	2690 (68.3%)	0.2452
Index Surgical Procedure			
Spine surgery	1225	1226	
Cesarean delivery	1154	1155	
Hysterectomy	155	155	
Total hip arthroplasty	616	617	
Total knee arthroplasty	789	793	
Tobacco use, N (%)			<0.0001
Ever	1251 (31.7%)	961 (24.4%)	
Never	2452 (62.1%)	2749 (69.8%)	
Unknown	236 (6.0%)	236 (6.0%)	
Alcohol use, N (%)			0.3599
Yes, N (%)	1153 (29.2%)	1172 (29.8%)	
No, N (%)	2468 (62.5%)	2489 (63.2%)	
Unknown, N (%)	318 (8.1%)	285 (7.2%)	
Any opioids over 180 days before index admission, N (%)	1014 (25.7%)	767 (19.5%)	<0.0001
Any opioids over 90 days before index admission, N (%)	648 (16.4%)	460 (11.7%)	<0.0001
Any opioids over 30 days before index admission, N (%)	383 (9.7%)	262 (6.7%)	<0.0001
180-day baseline ORAE—*w*/ or *w*/*o* opioid exposure, N (%)	967 (24.5%)	781 (19.8%)	<0.0001
Central nervous system disorder	40 (1.0%)	36 (0.9%)	0.6392
Cardiovascular	280 (7.1%)	252 (6.4%)	0.2011
Gastrointestinal	400 (10.1%)	320 (8.1%)	0.0016
Respiratory	213 (5.4%)	186 (4.7%)	0.1599
Skin	133 (3.4%)	92 (2.3%)	0.0053
Urinary	33 (0.8%)	14 (0.4%)	0.0053
Other	263 (6.7%)	135 (3.4%)	<0.0001
180-day baseline ORAE—among baseline opioid user, N (%)	373 (9.5%)	240 (6.1%)	<0.0001
Central nervous system disorder	21 (0.5%)	14 (0.4%)	0.2336
Cardiovascular	130 (3.3%)	77 (2.0%)	0.0002
Gastrointestinal	159 (4.0%)	97 (2.5%)	<0.0001
Respiratory	105 (2.7%)	68 (1.7%)	0.0043
Skin	42 (1.1%)	27 (0.7%)	0.0686
Urinary	18 (0.5%)	4 (0.1%)	0.0028
Other	128 (3.2%)	58 (1.5%)	<0.0001
Baseline Rx pain medication, N (%)	775 (19.6%)	727 (18.5%)	0.1571
Gabapentin	352 (8.9%)	373 (9.5%)	0.4276
Pregabalin	279 (7.1%)	228 (5.8%)	0.0182
Celecoxib	318 (8.1%)	273 (6.9%)	0.0515
Meloxicam	159 (4.0%)	122 (3.1%)	0.0236
Charlson Comorbidity Index, N (%)			
CCI 0	2795 (70.8%)	2808 (71.3%)	0.9823
CCI 1	688 (17.4%)	691 (17.5%)	
CCI 2	267 (6.8%)	265 (6.7%)	
CCI 3+	189 (4.8%)	182 (4.6%)	
Myocardial infarction	56 (1.4%)	59 (1.5%)	0.7854
Cerebrovascular disorder	90 (2.3%)	98 (2.5%)	0.5631
Congestive heart failure	87 (2.2%)	92 (2.3%)	0.7144
Diabetes, no complication	382 (9.7%)	401 (10.2%)	0.4906
Diabetes with complication	152 (3.9%)	179 (4.5%)	0.1337
Para/hemiplegia	34 (0.9%)	36 (0.9%)	0.816
Peptic ulcer	11 (0.3%)	14 (0.4%)	0.5508
Pulmonary disorder	491 (12.4%)	436 (11.1%)	0.051
Peripheral vascular disorder	101 (2.6%)	128 (3.2%)	0.0723
Renal disorder	111 (2.8%)	100 (2.5%)	0.435
Dementia	8 (0.2%)	12 (0.3%)	0.3726
Rheumatoid Arthritis	105 (2.7%)	95 (2.4%)	0.466
Liver, mild	85 (2.2%)	54 (1.4%)	0.0077
Liver, moderate to severe	15 (0.4%)	2 (0.1%)	0.0016
AIDS/HIV	13 (0.3%)	13 (0.3%)	0.9964
Cancer	0 (0.0%)	0 (0.0%)	n/a
Cancer, metastasis	0 (0.0%)	0 (0.0%)	n/a
Race/Ethnicity, N (%)			0.0100
Non-hispanic white	3184 (80.7%)	3071 (78.0%)	
Hispanic/Latino	389 (9.9%)	461 (11.7%)	
Other	335 (8.5%)	383 (9.7%)	
Unknown	31 (0.8%)	31 (0.8%)	
Index opioids, any, N (%)	3939 (99.8%)	3889 (98.7%)	<0.0001
Hydrocodone	415 (10.5%)	463 (11.8%)	0.0909
Buprenorphine	16 (0.4%)	0 (0.0%)	<0.0001
Butorphanol	0 (0.0%)	1 (0.0%)	0.3177
Codeine	5 (0.1%)	7 (0.2%)	0.5655
Fentanyl	1429 (36.2%)	1165 (29.6%)	<0.0001
Hydromorphone	2654 (67.3%)	1794 (45.5%)	<0.0001
Meperidine	404 (10.2%)	247 (6.3%)	<0.0001
Methadone	90 (2.3%)	31 (0.8%)	<0.0001
Morphine	1268 (32.1%)	1090 (27.7%)	<0.0001
Nalbuphine	249 (6.3%)	214 (5.4%)	0.0898
Oxycodone	3588 (90.9%)	3064 (77.8%)	<0.0001
Oxymorphone	2 (0.1%)	0 (0.0%)	0.2495
Pentazocine	0 (0.0%)	0 (0.0%)	n/a
Remifentanil	1733 (43.9%)	505 (12.8%)	<0.0001
Sufentanil	167 (4.2%)	179 (4.5%)	0.5203
Tapentadol	114 (2.9%)	111 (2.8%)	0.8287
Tramadol	1666 (42.2%)	1738 (44.1%)	0.1168
Anesthetic technique, N (%)			<0.0001
General	2147 (54.4%)	1578 (40.1%)	
Other	48 (1.2%)	39 (1.0%)	
Regional	1703 (43.2%)	2254 (57.2%)	
None	0 (0.0%)	1 (0.0%)	
Unknown	41 (1.0%)	74 (1.9%)	
Health plan, N (%)			<0.0001
Commercial	2096 (53.1%)	2041 (51.8%)	
Medicare	1069 (27.1%)	1326 (33.7%)	
Medicaid	574 (14.5%)	432 (11.0%)	
None	45 (1.1%)	32 (0.8%)	
Other	155 (3.9%)	115 (2.9%)	

BMI, body mass index; CCI, Charlson comorbidity index; IQR, interquartile range; LOS, length of stay; MME, morphine milligram equivalent; ORAE, opioid-related adverse event; SD, standard deviation.

**Table 2 healthcare-11-00115-t002:** Frequency and proportion of persistent opioid use: receiving 1+ opioid prescription between 31–180 days and 91–180 days after the index discharge.

Outcomes and Subgroup	High MME	Low MME	*p*-Value	
			Unadjusted	Holm
1+ Rx Opioid 31–180 days				
All, N (%)	1318 (33.5%)	964 (24.4%)	<0.0001	<0.0001
CD	91 (7.9%)	44 (3.8%)	<0.0001	<0.0001
UR	31 (20.0%)	21 (13.5%)	0.1285	0.1285
SS	520 (42.4%)	451 (36.8%)	0.0042	0.0042
THA	245 (39.8%)	141 (22.9%)	<0.0001	<0.0001
TKA	431 (54.6%)	307 (38.7%)	<0.0001	<0.0001
1+ Rx Opioid 91–180 days				
All, N (%)	771 (19.6%)	513 (13.0%)	<0.0001	<0.0001
CD	52 (4.5%)	23 (2.0%)	0.0007	0.0013
UR	14 (9.0%)	12 (7.7%)	0.6820	0.6820
SS	304 (24.8%)	262 (21.4%)	0.0430	0.0859
THA	153 (24.8%)	69 (11.2%)	<0.0001	<0.0001
TKA	248 (31.4%)	147 (18.5%)	<0.0001	<0.0001

Abbreviations: CD, cesarean delivery; THA, total hip arthroplasty; TKA, total knee arthroplasty; SS, spine surgery; UR, hysterectomy (uterus removal); Holm, *p*-values adjusted by Holm (Bonferroni step-down) method.

**Table 3 healthcare-11-00115-t003:** Patients who had one or more diagnosis of ORAE during the 180 days after index discharge.

Type of Surgery	ORAE Type	High MME *n* (%)	Low MME *n* (%)	*p*-Value	
				Unadjusted	Holm
All	Any	1073 (27.2%)	837 (21.2%)	<0.01	<0.01
	Central nervous system	111 (2.8%)	94 (2.4%)	0.22	0.45
	Cardiovascular	326 (8.3%)	219 (5.5%)	<0.01	<0.01
	Gastrointestinal	433 (11.0%)	332 (8.4%)	<0.01	<0.01
	Respiratory	276 (7.0%)	215 (5.4%)	<0.01	0.02
	Genitourinary	98 (2.5%)	63 (1.6%)	<0.01	0.02
	Skin	101 (2.6%)	85 (2.2%)	0.23	0.45
	Others	459 (11.7%)	310 (7.9%)	<0.01	<0.01
CD	Any	220 (19.1%)	138 (11.9%)	<0.01	<0.01
	Central nervous system	10 (0.9%)	6 (0.5%)	0.31	0.31
	Cardiovascular	70 (6.1%)	37 (3.2%)	<0.01	<0.01
	Gastrointestinal	97 (8.4%)	68 (5.9%)	0.02	0.06
	Respiratory	26 (2.3%)	8 (0.7%)	<0.01	0.01
	Genitourinary	8 (0.7%)	2 (0.2%)	0.06	0.11
	Skin	34 (2.9%)	13 (1.1%)	<0.01	0.01
	Others	67 (5.8%)	39 (3.4%)	<0.01	0.02
UR	Any	59 (38.1%)	47 (30.3%)	0.15	0.60
	Central nervous system	8 (5.2%)	1 (0.6%)	0.02	0.13
	Cardiovascular	22 (14.2%)	12 (7.7%)	0.07	0.41
	Gastrointestinal	37 (23.9%)	20 (12.9%)	0.01	0.10
	Respiratory	10 (6.5%)	11 (7.1%)	0.82	1.00
	Genitourinary	9 (5.8%)	3 (1.9%)	0.08	0.41
	Skin	4 (2.6%)	7 (4.5%)	0.36	1.00
	Others	23 (14.8%)	25 (16.1%)	0.75	1.00
SS	Any	390 (31.8%)	328 (26.8%)	<0.01	0.04
	Central nervous system	69 (5.6%)	62 (5.1%)	0.53	0.85
	Cardiovascular	126 (10.3%)	91 (7.4%)	0.01	0.08
	Gastrointestinal	167 (13.6%)	142 (11.6%)	0.13	0.51
	Respiratory	108 (8.8%)	92 (7.5%)	0.24	0.71
	Genitourinary	55 (4.5%)	38 (3.1%)	0.07	0.36
	Skin	26 (2.1%)	32 (2.6%)	0.43	0.85
	Others	188 (15.3%)	130 (10.6%)	<0.01	<0.01
THA	Any	156 (25.3%)	128 (20.7%)	0.06	0.38
	Central nervous system	9 (1.5%)	7 (1.1%)	0.61	1.00
	Cardiovascular	32 (5.2%)	31 (5.0%)	0.89	1.00
	Gastrointestinal	49 (8.0%)	41 (6.6%)	0.38	1.00
	Respiratory	45 (7.3%)	29 (4.7%)	0.05	0.38
	Genitourinary	14 (2.3%)	8 (1.3%)	0.20	0.98
	Skin	17 (2.8%)	12 (1.9%)	0.35	1.00
	Others	73 (11.9%)	52 (8.4%)	0.05	0.37
TKA	Any	248 (31.4%)	196 (24.7%)	<0.01	0.02
	Central nervous system	15 (1.9%)	18 (2.3%)	0.61	1.00
	Cardiovascular	76 (9.6%)	48 (6.1%)	<0.01	0.05
	Gastrointestinal	83 (10.5%)	61 (7.7%)	0.05	0.25
	Respiratory	87 (11.0%)	75 (9.5%)	0.30	1.00
	Genitourinary	12 (1.5%)	12 (1.5%)	0.99	1.00
	Skin	20 (2.5%)	21 (2.6%)	0.89	1.00
	Others	108 (13.7%)	64 (8.1%)	<0.01	<0.01

Abbreviations: CD, cesarean delivery; THA, total hip arthroplasty; TKA, total knee arthroplasty; SS, spine surgery; UR, hysterectomy (uterus removal); Holm, *p*-values adjusted by Holm (Bonferroni step-down) method.

**Table 4 healthcare-11-00115-t004:** Healthcare resource utilization: number of admissions, cumulative length of inpatient stay, cumulative length of inpatient stay if patient was admitted ever, average length of inpatient stay per admission during 180-day post-discharge period.

Type of Surgery (*n*, High MME vs. Low MME)	Mean (SD)				Median [IQR]; Min–Max			
	High MME	Low MME	*p*-Value *		High MME	Low MME	*p*-Value ^†^	
			Unadjusted	Holm			Unadjusted	Holm
Number of readmissions								
All (3939 vs. 3946)	0.18 (0.51)	0.13 (0.44)	<0.01	<0.01	0 [0–0]; 0–5	0 [0–0]; 0–7	<0.01	<0.01
CD (1154 vs. 1155)	0.05 (0.25)	0.02 (0.15)	<0.01	<0.01	0 [0–0]; 0–3	0 [0–0]; 0–2	<0.01	<0.01
UR (155 vs. 155)	0.12 (0.45)	0.11 (0.43)	0.80	1.00	0 [0–0]; 0–4	0 [0–0]; 0–3	0.57	1.00
SS (1225 vs. 1226)	0.22 (0.59)	0.18 (0.57)	0.05	0.36	0 [0–0]; 0–5	0 [0–0]; 0–7	<0.01	0.04
THA (616 vs. 617)	0.29 (0.66)	0.15 (0.40)	<0.01	<0.01	0 [0–0]; 0–5	0 [0–0]; 0–3	<0.01	<0.01
TKA (789 vs. 793)	0.25 (0.52)	0.20 (0.50)	0.04	0.36	0 [0–0]; 0–3	0 [0–0]; 0–5	0.02	0.12
Cumulative length of stay								
All (3939 vs. 3946)	1.09 (4.75)	0.64 (3.56)	<0.01	<0.01	0 [0–0]; 0–109	0 [0–0]; 0–109	<0.01	<0.01
CD (1154 vs. 1155)	0.20 (1.30)	0.07 (0.73)	<0.01	0.03	0 [0–0]; 0–26	0 [0–0]; 0–16	<0.01	<0.01
UR (155 vs. 155)	0.94 (6.86)	0.70 (2.98)	0.70	1.00	0 [0–0]; 0–83	0 [0–0]; 0–21	0.62	1.00
SS (1225 vs. 1226)	1.87 (7.21)	1.17 (5.53)	<0.01	0.06	0 [0–0]; 0–109	0 [0–0]; 0–109	<0.01	0.02
THA (616 vs. 617)	1.29 (3.64)	0.55 (2.30)	<0.01	<0.01	0 [0–0]; 0–41	0 [0–0]; 0–42	<0.01	<0.01
TKA (789 vs. 793)	1.08 (2.84)	0.70 (2.89)	<0.01	0.07	0 [0–0]; 0–32	0 [0–0]; 0–45	<0.01	0.02
Cumulative length of stay, among ever admitted								
All (561 vs. 399)	7.6 (10.41)	6.3 (9.5)	0.54	1.00	4 [3–8]; 1–109	4 [2–6]; 1–109	0.21	1.00
CD (45 vs. 19)	5.1 (4.38)	4.3 (3.9)	0.6	1.00	4 [2–6]; 1–26	3 [2–4]; 2–16	0.68	1.00
UR (15 vs. 12)	9.7 (20.67)	9.1 (6.4)	0.63	1.00	4 [2–5]; 1–83	7.5 [4.5–13]; 2–21	0.50	1.00
SS (203 vs. 153)	11.3 (14.4)	9.4 (13.0)	0.48	1.00	6 [3–11]; 1–109	5 [4–10]; 1–109	0.50	1.00
THA (130 vs. 84)	6.1 (5.8)	4.1 (5.0)	<0.01	<0.01	4 [3–8]; 1–41	3 [2–4]; 1–42	<0.01	<0.01
TKA (168 vs. 131)	5.1 (4.2)	4.2 (6.0)	0.72	1.00	4 [3–6]; 2–32	2 [2–4]; 1–45	0.93	1.00
Average length of inpatient stay per admission, among ever admitted								
All (561 vs. 399)	5.4 (5.6)	4.4 (3.7)	<0.01	<0.01	4 [3–6]; 1–55	3 [2–5]; 1–31	<0.01	<0.01
CD (45 vs. 19)	4.0 (2.7)	3.6 (3.2)	0.61	1.00	3 [2–5]; 1–14	3 [2–3.5]; 2–16	0.27	1.00
UR (15 vs. 12)	5.1 (5.3)	6.5 (3.9)	0.44	1.00	4 [2–5]; 1–20.75	5.7 [4.25–7.5]; 2–17	0.08	0.65
SS (203 vs. 153)	7.7 (8.2)	6.0 (4.8)	0.02	0.12	5.5 [3–8]; 1–54.5	5 [3–7]; 1–31	0.06	0.45
THA (130 vs. 84)	4.1 (2.3)	3.4 (2.3)	0.03	0.23	3.25 [3–5]; 1–14	3 [2–4]; 1–14	<0.01	<0.01
TKA (168 vs. 131)	4.2 (2.1)	3.1 (1.7)	<0.01	<0.01	4 [3–5]; 2–14	2 [2–4]; 1–10.75	<0.01	<0.01

* Student *t*-test, ^†^ Wilcoxon signed-rank test. Abbreviations: CD, cesarean delivery; THA, total hip arthroplasty; TKA, total knee arthroplasty; SS, spine surgery; UR, hysterectomy (uterus removal), Holm, *p*-values adjusted by Holm (Bonferroni step-down) method.

**Table 5 healthcare-11-00115-t005:** Healthcare resource utilization: number of office visits and ED visits during 180-day post-discharge period and number of telephone encounters during 7-day post discharge period.

Type of Surgery (*n*, High MME vs. Low MME)	Mean (SD)				Median [IQR]; Min–Max			
	High MME	Low MME	*p*-Value *		High MME	Low MME	*p*-Value ^†^	
			Unadjusted	Holm			Unadjusted	Holm
Number of office visits								
All (3939 vs. 3946)	5.36 (6.91)	5.05 (6.23)	0.04	0.29	3 [1–7]; 0–109	3 [1–6]; 0–67	0.03	0.21
CD (1154 vs. 1155)	1.89 (2.70)	1.49 (2.13)	<0.01	<0.01	1 [0–3]; 0–34	1 [0–2]; 0–21	<0.01	0.01
UR (155 vs. 155)	3.26 (4.53)	2.23 (3.08)	0.02	0.16	2 [0–5]; 0–32	1 [0–3]; 0–18	0.10	0.83
SS (1225 vs. 1226)	6.15 (8.00)	5.64 (6.30)	0.08	0.64	4 [2–7]; 0–109	4 [2–7]; 0–67	0.59	1.00
THA (616 vs. 617)	6.29 (5.74)	6.21 (5.85)	0.82	1.00	4 [2–8]; 0–49	4 [2–8]; 0–40	0.40	1.00
TKA (789 vs. 793)	8.88 (8.13)	8.94 (7.76)	0.88	1.00	6 [3–12]; 0–51	6 [3–13]; 0–47	0.63	1.00
Number of ER visits								
All (3939 vs. 3946)	0.24 (0.86)	0.18 (0.91)	<0.01	0.08	0 [0–0]; 0–16	0 [0–0]; 0–32	<0.01	<0.01
CD (1154 vs. 1155)	0.24 (0.80)	0.16 (0.53)	<0.01	0.03	0 [0–0]; 0–12	0 [0–0]; 0–6	0.03	0.26
UR (155 vs. 155)	0.35 (1.09)	0.05 (0.26)	<0.01	<0.01	0 [0–0]; 0–10	0 [0–0]; 0–2	<0.01	<0.01
SS (1225 vs. 1226)	0.24 (0.92)	0.30 (1.44)	0.18	1.00	0 [0–0]; 0–16	0 [0–0]; 0–32	0.40	1.00
THA (616 vs. 617)	0.22 (0.85)	0.10 (0.48)	<0.01	0.03	0 [0–0]; 0–10	0 [0–0]; 0–5	<0.01	<0.01
TKA (789 vs. 793)	0.22 (0.78)	0.14 (0.53)	<0.01	0.08	0 [0–0]; 0–8	0 [0–0]; 0–7	0.05	0.41
Number of telephone encounters								
All (3939 vs. 3946)	1.28 (1.63)	1.09 (1.42)	<0.01	<0.01	1 [0–2]; 0–11	1 [0–2]; 0–11	<0.01	<0.01
CD (1154 vs. 1155)	0.46 (0.78)	0.33 (0.68)	<0.01	<0.01	0 [0–1]; 0–5	0 [0–0]; 0–4	<0.01	<0.01
UR (155 vs. 155)	1.17 (1.32)	1.52 (1.27)	0.02	0.15	1 [0–2]; 0–6	1 [1–2]; 0–5	<0.01	0.04
SS (1225 vs. 1226)	1.14 (1.19)	1.16 (1.12)	0.72	1.00	1 [0–2]; 0–9	1 [0–2]; 0–8	0.24	1.00
THA (616 vs. 617)	2.00 (2.09)	1.40 (1.71)	<0.01	<0.01	1 [0–3]; 0–10	1 [0–2]; 0–9	<0.01	<0.01
TKA (789 vs. 793)	2.13 (2.08)	1.76 (1.86)	<0.01	<0.01	2 [1–3]; 0–11	1 [0–2]; 0–11	<0.01	<0.01

* Student *t*-test, ^†^ Wilcoxon signed-rank test. Abbreviations: CD, cesarean delivery; THA, total hip arthroplasty; TKA, total knee arthroplasty; SS, spine surgery; UR, hysterectomy (uterus removal); Holm, *p*-values adjusted by Holm (Bonferroni step-down) method.

## Data Availability

The data presented in this study are available upon reasonable request from the corresponding author. The research data are not publicly available due to the privacy and confidentiality.

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
