# Peer review of "Positive Association between Peri-Surgical Opioid Exposure and Post-Discharge Opioid-Related Outcomes"

_healthcare, 2022, doi:10.3390/healthcare11010115_

Round 1

Reviewer 1 Report

Summary:

The submitted manuscript evaluates the association between pre/peri-surgical opioid use and post-surgical outcomes. Outcomes are defined over key metrics like healthcare resource utilization, and opioid prescription usage.

Comments:

1. line 14-27 - Are these in a different font from the other text in the manuscript?

2. ORAE/ORADE abbreviations have been used without specifying what they stand for. Please include the expanded form at the first instance of use of these abbreviations.

3. Table1 High MME IQR is 92.7-66.67. This might be a typo.

4. line 121-123 -"The purpose of this study was to determine any signal in increasing ORAEs associated with index MME, and we did not specifically determine if the diagnosis of ORAEs was caused by opioid use"

Could the authors please elaborate on this point? Does this mean that ORAE can't be used as an outcome metric, because the ORAE could be caused by other unrelated phenomenon?

4. Could the authors also elaborate on which hospital admissions were considered as part of the outcomes? Are planned and unplanned admissions treated the same?

5. Table1 , the number of participants in Low MME keeps increasing year-to-year. Is this an indication of changing opioid prescribing policies?

6. The authors mention in Line 136 that they built a multivariable logistic regression model after performing variable selection. I'm unable to determine which table contains the results of this model. Please include the results of your model as a table(supplementary is fine).

Suggestions:

My biggest concern is the various comparisons being without correcting for multiple-testing bias.

I would recommend that the authors consider either Bonferroni Step-Down or Westfall and Young Permutation corrections when evaluating significance. 

The authors can determine whether they prefer false-positives or false-negatives and select the appropriate correction methods, but some correction is a must.

Author Response

Comments and Suggestions for Authors

Summary:

The submitted manuscript evaluates the association between pre/peri-surgical opioid use and post-surgical outcomes. Outcomes are defined over key metrics like healthcare resource utilization, and opioid prescription usage.

Comments:

1. line 14-27 - Are these in a different font from the other text in the manuscript?

RE) We have checked the font and could not find any difference in formatting/font. This probably happened while we completed the online submission. We will ensure that formats are consistent across the manuscript during the proofread once the study has been accepted.

2. ORAE/ORADE abbreviations have been used without specifying what they stand for. Please include the expanded form at the first instance of use of these abbreviations.

RE) We addressed this comments in the 3rd page of the main text, 3rd paragraph of the Methods section, ORAE stands for opioid-related adverse event. ORADE was replaced with ORAE throughout the revised manuscript.

3. Table1 High MME IQR is 92.7-66.67. This might be a typo.

RE) Thanks for pointing this error out. The numbers had been flipped between High MME and Low MME group. We made the correction

4. line 121-123 -"The purpose of this study was to determine any signal in increasing ORAEs associated with index MME, and we did not specifically determine if the diagnosis of ORAEs was caused by opioid use"

Could the authors please elaborate on this point? Does this mean that ORAE can't be used as an outcome metric, because the ORAE could be caused by other unrelated phenomenon?

RE) We adjust for the presence of ORAE diagnosis during the baseline period using a multivariable regression model. This, we believe, address some of your concern.

In a causal inference frame, we acknowledge that this multivariable approach provide us with limited knowledge. We have discussed this in the discussion.

4. Could the authors also elaborate on which hospital admissions were considered as part of the outcomes? Are planned and unplanned admissions treated the same?

RE) In the end of the “outcomes” section, we disclosed that resource utilization as secondary outcomes are hospital readmissions, length of stay, number of office visits and emergency department visits for any reason. Thus, the outcomes include both planned and unplanned admissions.

Not being approved for free-text record use, we cannot accurately determine if the admission was planned or unplanned. This is one of the general limitations in conducting a retrospective observational research using structured data, and is not necessary to be specifically discussed.  

5. Table1 , the number of participants in Low MME keeps increasing year-to-year. Is this an indication of changing opioid prescribing policies?

RE) We are unaware of policies that specifically influenced the MME. The variables listed in the same paragraph were selected based on the step-wise model selection process. We edited the text to clarify.

6. The authors mention in Line 136 that they built a multivariable logistic regression model after performing variable selection. I'm unable to determine which table contains the results of this model. Please include the results of your model as a table (supplementary is fine).

RE) The variables listed in the same paragraph were selected based on the step-wise model selection process. We edited the text to clarify. Adjusted odds ratios are presented in the figure (Forest plots).

Suggestions:

My biggest concern is the various comparisons being without correcting for multiple-testing bias.

I would recommend that the authors consider either Bonferroni Step-Down or Westfall and Young Permutation corrections when evaluating significance.

The authors can determine whether they prefer false-positives or false-negatives and select the appropriate correction methods, but some correction is a must.

RE) We corrected the p-values from bivariate analysis for each set of outcomes: primary outcome, ORAE and HCRU using the Bonferroni step-down approach (i.e., Holms method). While we adjust the p-values, we also corrected several p-values that had been copy-pasted incorrectly. The Holms method correction has a nominal to moderate impact on statistical inference.

Reviewer 2 Report

The manuscript titled: “Positive association between peri-surgical opioid exposure and post-discharge opioid related outcomes” aimed at investigating the association between peri-surgical opioid use and post-discharge outcomes, including persistent postsurgical opioid prescription, opioid-related symptoms and healthcare resource utilization.

The work is interesting; however it has been submitted in the Section “Coronaviruses (CoV) and COVID-19 Pandemic”, within the Special Issue on "Opioid Crisis during the COVID-19 Pandemic". As reported, this Special Issue aims to seek out the most contemporary evidence-based studies considering health utilization and policy of opioids, medication-assisted treatment, and cancer treatment in the COVID-19 era.

I am sorry to tell you that your analysis based on data from an academic healthcare system between 2015 and 2018 is not adding any information about opioid use and relative healthcare resource utilization during COVID-19 pandemic.

My suggestion is therefore to extend the temporal window and cover (at least) the period of the first waves of COVID-19. Importantly, this will allow to study and compare the use of healthcare resources due to opioid exposure before and after the COVID-19 outbreak.

Only after this substantial change the paper can be reconsider for publication.

Author Response

The work is interesting; however it has been submitted in the Section “Coronaviruses (CoV) and COVID-19 Pandemic”, within the Special Issue on "Opioid Crisis during the COVID-19 Pandemic". As reported, this Special Issue aims to seek out the most contemporary evidence-based studies considering health utilization and policy of opioids, medication-assisted treatment, and cancer treatment in the COVID-19 era.

I am sorry to tell you that your analysis based on data from an academic healthcare system between 2015 and 2018 is not adding any information about opioid use and relative healthcare resource utilization during COVID-19 pandemic.

My suggestion is therefore to extend the temporal window and cover (at least) the period of the first waves of COVID-19. Importantly, this will allow to study and compare the use of healthcare resources due to opioid exposure before and after the COVID-19 outbreak.

Only after this substantial change the paper can be reconsider for publication.

RE) Thank you so much for the comments regarding the inclusion of the data from COVID-19 outbreak period. Rather unfortunately, the investigators changed their affiliations and currently have limited ability to abstract additional study data from the data source. Further, the COVID-19 period would significantly influence opioid utilization and healthcare resource use, which is not within the scope of the originally proposed and approved research.

We understand that the study does not cover the COVID-19 outbreak era. If the contents are not acceptable for the current issue but are of interest to MDPI Healthcare, proceeding with our manuscript for the following volume or issue is an acceptable alternative for the investigators. However, by pointing out the questions on perisurgical opioid use followed by persistent opioid use that has been underinvestigated throughout the pre-COVID and COVID era, we still believe that our study can support this special issue. This, we believe, is why we were recommended by the editorial board/special issue editors to submit our research under the current issue. Thanks for your consideration.

Round 2

Reviewer 2 Report

Dear authors,

Please add a sentence in the discusson session to emphasize the importance of pointing out the questions on perisurgical opioid use followed by persistent opioid use that has been underinvestigated throughout the pre-COVID and, most likely, the COVID era.

Author Response

In our revision, we specifically discuss the positive association between the perisurgical opioid and post-discharge outcomes as a perpetual issue across the study period and COVID pandemic. In the limitation, we pointed out that future effort is warranted to extend the study period to cover the COVID epidemic and adjust for the pre-/post- pandemic effects